# Inoculation with indigenous nitrogen-fixers enhances seedling growth and nutrient uptake in a greenhouse bioassay

**Majda K. Suleiman**[☉], **Ali M. Quoreshi**[iD][☉][*], **Anitha J. Manuvel**[‡], **Mini T. Sivadasan**[‡], **Sheena Jacob**[iD][‡]

Food Security Program, Environment and Life Sciences Research Center, Kuwait Institute for Scientific Research, Kuwait

☉ These authors contributed equally to this work
‡ These authors also contributed equally to this work
* aquoreshi@kisr.edu.kw

## Abstract

Desert ecosystems in Kuwait are increasingly affected by land degradation, resulting in nutrient-limited soils that constrain native plant establishment. Harnessing indigenous diazotrophic bacteria adapted to arid environments may provide a sustainable strategy to improve plant growth and nutrient acquisition. Free-living and root-associated nitrogen-fixing bacteria contribute substantially to nitrogen inputs in arid ecosystems and may enhance plant growth, performance and nutrient acquisition under nutrient-poor conditions. This study evaluated the growth performance and nutrient uptake ability of four native plant species of Kuwait following inoculation with a consortium of selected indigenous putative diazotrophs isolated from the Kuwait desert soils. The seedlings of *Vachellia pachyceras* were inoculated with both indigenous root-nodule bacteria isolated from Kuwait desert and a commercial inoculum to evaluate their symbiotic efficiency. The seedlings were cultivated under greenhouse conditions using either native desert soils or a potting mix substrate to assess the influence of growth medium or inoculation response. Across species, inoculation significantly enhanced plant dry mass and nutrient uptake compared to the non-inoculated controls. The magnitude of improvement varied among bacterial density, host plants, and growth substrate. These findings support the potential use of indigenous diazotrophs as biofertilizers to enhance plant growth and nutrient uptake of native plant species, and for restoration and revegetation efforts in arid environments. However, direct measurements of nitrogen fixation were not conducted and should be addressed in future field-based studies. This study represents the first evaluation of Kuwait's native seedlings inoculated with indigenous diazotrophs, highlighting their potential for sustainable ecosystem restoration.

**Data availability statement:** All relevant data are within the paper.

**Funding:** This study was funded by Kuwait Foundation for the Advancement of Sciences (KFAS). The grand number is Project – P215-42SL-01 The funders had no role in study design, data collection and analysis, decision to publish, or preparation of the manuscript.

**Competing interests:** The authors have declared that no competing interests exist.

## Introduction

Land degradation in the Kuwait desert, driven by environmental extremes and anthropogenic disturbances, continues to accelerate desertification, soil degradation, and decline of native vegetation [1,2]. These pressures reduce soil fertility, disrupt plant-microbe interactions, and ultimately limit natural ecosystem recovery. Evidences suggest that the native plants' population in Kuwait has been diminishing in recent years due to soil degradation, extreme weather, climatic fluctuations and various anthropogenic activities [1,2].

Nevertheless, desert indigenous plants are adapted to the extreme local environmental conditions, and characterized by slower growth and adaptation mechanisms that enable them to survive in harsh environments. Desert soils in Kuwait are typically characterized by low organic matter, limited nitrogen availability, and poor microbial activity, all of which constrain successful revegetation efforts [3]. Successful restoration of native plant communities requires recovery of soil fertility and rhizosphere microbial communities, as the soil is the primary source of nutrients for plant growth and development. Although desert environments are nutrient-limited, they host diverse and resilient microbial communities that exist as free-living, associate with plants, or components of soil biocrusts [4]. The soil microbial communities are considered key driving force for regulating critical soil ecosystems processes, including organic matter decomposition and nutrient cycling, nutrient mineralization, and biogeochemical cycling of terrestrial ecosystems [5–8]. In Kuwait's desert soil, nutrient availability for native plant is severely deficient due to very low organic matter (<1%), low clay content, high calcareous materials, and deficiencies in essential nutrients [9]. Nutrient availability is therefore a critical constraint in restoring degraded desert soils. Under extreme arid environmental conditions, nutrient biogeochemical cycling, especially N cycling is largely mediated by microorganisms [10]. Nitrogen-fixing bacteria and other soil microbes could play an important role in nutrient cycling, decomposition of organic matter, and improvement of soil fertility, and maintenance of healthy ecosystems [11]. To plan effective strategies for conserving and restoring desert ecosystems, it is often necessary to manipulate and establish associated soil microbial communities to support sustainable vegetation cover.

Diazotrophs are rhizospheric or root-associated bacteria capable of fixing atmospheric dinitrogen into plant-available forms, thereby improving soil fertility in nutrient-poor soils and enhance the accessibility of nitrogen to the host plant [12–14]. In arid systems, indigenous diazotrophic communities are of particular interest because they are naturally adapted to high temperature, salinity, and moisture stress. Their application as bioinoculants may provide a cost-effective and environmentally sustainable approach to enhance native plant establishment in restoration program. Studies by Bashan et al. [15] and Moreno et al. [16] demonstrated that plant-growth-promoting bacteria (PGPB) enhance native plant growth and improve soil stability in degraded forest and desert soils. Similarly, Liu et al. [17] reported accelerated plant development and higher survival rate of different forest seedlings following bacterial inoculation. In particular, plant-growth-promoting (PGP) nitrogen-fixing bacteria offers a suitable alternative to chemical fertilizers

[18], whose excessive use raises environmental and economic concerns. In view of the growing interest in developing alternative approaches to improve soil fertility, the use of bioinoculants and biofertilizers containing plant-beneficial microorganisms have gained attention as promising alternative to chemical fertilizers and potential strategies to enhance soil fertility by mobilizing the essential nutrients to the plants [19]. In Kuwait's desert soils, limited nitrogen availability underscores the necessity to develop a strategy to enhance nutrient supply through diazotrophic bacteria to improve the soil fertility [20]. Although numerous studies have documented the significant impact of plant growth-promoting bacteria (PGPB) and other beneficial microbes on crop yield and productivity, their role in promoting growth and biomass of desert seedlings through free-living or root-associated beneficial microorganisms remains insufficiently explored [21]. In Kuwait, the application of indigenous free-living diazotrophs and root-associated PGPB to support native plant growth has not been investigated.

Previous work in Kuwait deserts isolated multiple putative diazotrophic strains from rhizosphere soils and root nodules of native plants. In that study, several free-living and symbiotic nitrogen fixing were identified based on 16S rRNA sequencing [20]). However, their functional performance on native host plants under controlled conditions has not been comprehensively evaluated. The hypothesis of this research was that inoculation of free-living and root-associated diazotrophs may increase growth and nutrient uptake of native desert plants under the current experimental setting. Therefore, the present study aimed to assess the effect of a consortium of indigenous diazotrophs obtained from our previous study could contribute on early seedling development and nutrient acquisition of selected native desert plant species under controlled greenhouse conditions, thereby supporting their potential use as biofertilizer in future large-scale restoration and revegetation efforts.

## Materials and methods

### Inoculum preparation

The bacterial isolates used in this greenhouse experiment are indigenous diazotrophic strains isolated from the rhizospheric soil of *Rhanterium epapposum*, *Farsetia aegyptia*, and *Haloxylon salicornicum* (free-living nitrogen-fixing bacteria), as well as root nodules of *Vachellia pachyceras* (symbiotic nitrogen-fixers) obtained from our previous study. The nitrogen-fixing capacity of isolated diazotrophs was previously tested and confirmed using the Acetylene Reduction Assay, and they were identified using the 16s rRNA gene sequencing method [20]. Totally, 6, 3, 11 and 19 strain of indigenous nitrogen-fixers were used as inoculum to test on the native plant species *R. epapposum, F. aegyptia, H. salicornicum*, and *V. pachyceras*, respectively. The taxonomic identity and source of each isolate are summarized in Supplementary S1 Table. For each bacterial isolate obtained from a single plant species, cell suspensions were prepared in appropriate broth media at a density of $10^8$ CFU/mL [22,23]. These isolates were then pooled together to form a mixture of bacterial inoculum with a final cell density of $10^8$ CFU/mL. From this mixture, a single dilution of $10^4$ CFU/mL was prepared for *F. aegyptia, R. epapposum*, and *H. salicornicum* (free-living nitrogen-fixing bacteria). For *V. pachyceras*, a dilution series of $10^8$, $10^6$, $10^4$, and $10^2$ CFU/mL was prepared.

### Seedling Production

Two types of seedling growth media (native desert soil and commercial potting mix) were used in the experiment. Desert soil was collected from KISR's Station for Research and Innovation (KSRI), packed in double autoclavable bags, and transported to the laboratory. The physiochemical properties of the experimental soil are presented in Supplementary S2 Table. Potting mix soil was prepared by mixing agricultural soil, peat moss, and potting soil in a 2:1:1 volume-to-volume ratio (v/v) and packed into double autoclavable bags. The packed soils were sterilized twice in an autoclave at 121°C for 30 minutes, with complete cooling and mixing between cycles, and then air-dried. Likewise, jiffy pots and potting mix soil used in the initial seedling production were also sterilized prior to use. The irrigation water was sterilized once in

an autoclave at 121°C for 15 min and used throughout the experiment. The sterilized jiffy pots were filled with autoclaved potting mix soil or desert soil and saturated with sterile water, and placed in a tray (30 jiffy pots). As Kuwait's native plant seeds were very sensitive to chemical treatment, the surface sterilization was done by soaking *R. epapposum* and *F. aegyptia* overnight in sterile water and washed six times in sterile water, and one capitulum of *R. epapposum* was sown on the surface of the sterile potting mix soil or desert soil in jiffy pots. The seeds of *F. aegyptia* were sown in jiffy pots filled with the potting mix or desert soil for germination. *H. salicornicum* seeds were washed six times with sterile water, and one seed was sowed on the top of the potting mix soil in jiffy pot for germination. *Vachellia pachyceras* seeds were treated with concentrated sulphuric acid for 30 min [24], washed six times in sterile water, and soaked in sterile water overnight. Sulphuric acid scarification was applied only to *V. pachyceras* because its seed possess a hard, water-impermeable coat that requires chemical scarification to ensure uniform germination by breaking physical dormancy. The imbibed seeds were transferred to sterile petri-plate with moistened filter paper until germination. The germinated seeds were transferred to sterile jiffy pots with potting soil mix or desert sand. All the trays were maintained in the standard greenhouse conditions (25±2°C, 60–70% relative humidity, and a 14-h photoperiod) and irrigated with sterile water throughout the experiment as required.

### Transplantation and inoculation

Two-weeks after the germination in the jiffy pot, the seedlings of all the species were transplanted to one-gallon pots and maintained in the greenhouse. Six-weeks after the transplantation, the seedlings were inoculated with 20 ml of indigenous bacterial suspension with desired cell densities ($10^4$ and $10^8$ CFU/mL for *R. epapposum, F. aegyptia, H. salicornicum* whereas $10^2$, $10^4$, $10^6$ $10^8$ CFU/mL for *V. pachyceras*) [25,26]. Seedlings of all the species in the control treatment were not inoculated. Each treatment including control had eight seedlings as replicates which were arranged in Completely Randomized Design (CRD) in the greenhouse conditions. Furthermore, for comparative purposes, *V. pachyceras* was inoculated with 10 ml of 1 OD commercial strains, American Type Culture Collection (ATCC) *Rhizobium leguminosarum* (ATCC®10004™) and *Bradyrhizobium* sp. (ATCC®BAA-1182) as reference inoculants along with the main experiment, to compare the potential of commercial inoculum to form nodulation and improve nitrogen uptake of *V. pachyceras* under greenhouse conditions.

### Data collection and data analysis

The inoculated and non-inoculated control seedlings were maintained under greenhouse conditions and their growth performance (plant height, number of leaves, shoot and root biomass, stem diameter) and physical condition (vigor) were recorded before the experiment was terminated after the 10th month. At the end of the experiment, five randomly selected seedlings per treatment were excavated completely, and the soil was rinsed off the roots. The number and weight of the root nodules of *V. pachyceras* were recorded. The seedlings were separated into roots and shoots and dried in an oven at 70°C for 48 hr. or until a constant dry weight was achieved. The dry mass of roots and shoots of each plant species was determined on an analytical scale. The dried powdered shoot samples of each plant species were analyzed for Nitrogen (N), Phosphorus (P), and Potassium (K) concentrations using standard laboratory procedures.

Seedling biomass, plant parameters and chemical analysis data were analyzed using Analysis of Variance Procedure (ANOVA) using SPSS® software – version 22 (IBM®) and treatment means were compared using the Duncan's Multiple Range test to ascertain the significant differences among treatments at $P < 0.05$. level of probability [27]. Prior to analysis, data normality was verified using Shapiro-Wilk test and homogeneity of variances using Levene's Test. Type of growth medium and bacterial inoculum were considered as factors in the analysis.

### Results

Effect of potential nitrogen-fixer bacterial inoculations on plant growth parameters of selected native plant species

### R. epapposum

Generally, inoculation with the two densities ($10^4$ and $10^8$) of indigenous bacterial inoculum to *R. epapposum* significantly enhanced the average root (p<0.001), shoot (p<0.001), and total biomass (p<0.001) per plant compared to the non-inoculated control (Table 1). Additionally, average root biomass per plant increased significantly (p<0.001) in plants grown in potting soil when compared to that grown in desert soil (Table 1). However, bacterial inoculation and effect of growth medium types used had no statistically significant impact on any other growth parameters determined when compared to the control (Table 1). The average root biomass of the seedlings grown in desert soil medium increased by 88% when compared to the seedlings grown in potting mix soil (Fig 1). Likewise, the indigenous bacterial inoculation increased average root biomass by 331% and 538%, average shoot biomass by 198% and 303% and average total biomass by 206% and 318% when inoculated with $10^4$ CFU/mL and $10^8$ CFU/mL densities (Fig 1). Although there was significant difference in the average root biomass and shoot biomass between non-inoculated control and those inoculated with indigenous bacteria, there was no significant difference within the two inoculum densities used ($10^4$ CFU/mL and $10^8$ CFU/mL) (Fig 1). Unlike the average root and shoot biomass, the average total biomass was significantly affected by inoculum density, which resulted in 54% increase in average total biomass of seedlings inoculated with indigenous inoculum of density $10^8$ cells compared to those inoculated with cell density of $10^4$ (Fig 1).

### F. aegyptia

*F. aegyptia*, seedlings inoculated with the indigenous bacterial inoculum at $10^8$ cells exhibited significantly greater root (P≤0.005), shoot (P≤0.001), and total biomass (P≤0.001) compared to those inoculated with $10^4$ cells and the control (Table 1). Bacterial inoculation did not show statistically significant differences on other growth parameters.

Interestingly, plant height (P≤0.016), stem diameter (P≤0.026), shoot biomass (P≤0.001) and total biomass (P<0.001) of seedlings grown in desert soil was significantly higher than those grown in potting mix. However, soil medium did not exhibit statistically significant effect on other growth parameters (Table 1).

**Table 1. Results of Analysis of Variance (P values) Testing for the Effect of Growth Medium and Bacterial Inoculum on Growth Parameters of Selected Native Plant Species.**

| Variable | Plant Height | Stem Diameter | Plant Vigor | Root Biomass | Shoot Biomass | Total Biomass | Number of Root Nodules | Weight of Nodules | Root Shoot Ratio |
|---|---|---|---|---|---|---|---|---|---|
| *Rhanterium epapposum* | | | | | | | | | |
| Soil Media | 0.499 | 0.386 | 0.584 | 0.011 | 0.496 | 0.769 | NA | NA | 0.150 |
| Bacterial Inoculation | 0.118 | 0.285 | 0.510 | <0.001 | <0.001 | <0.001 | NA | NA | 0.894 |
| Soil Media * Bacterial Inoculation | 0.243 | 0.413 | 0.114 | 0.054 | 0.725 | 0.677 | NA | NA | 0.585 |
| *Farsetia aegyptia* | | | | | | | | | |
| Soil Media | 0.016 | 0.026 | 0.170 | 0.065 | 0.001 | <0.001 | NA | NA | 0.546 |
| Bacterial Inoculation | 0.453 | 0.795 | 0.613 | 0.005 | <0.001 | <0.001 | NA | NA | 0.244 |
| Soil Media * Bacterial Inoculation | 0.194 | 0.413 | 0.613 | 0.987 | 0.335 | 0.332 | NA | NA | 0.872 |
| *Haloxylon salicornicum* | | | | | | | | | |
| Soil Media | 0.082 | 0.646 | 1.000 | 0.007 | 0.001 | <0.001 | NA | NA | 0.009 |
| Bacterial Inoculation | 0.373 | 0.653 | 0.613 | 0.416 | <0.001 | <0.001 | NA | NA | 0.961 |
| Soil Media * Bacterial Inoculation | 0.138 | 0.414 | 0.243 | 0.428 | 0.046 | 0.050 | NA | NA | 0.907 |
| *Vachellia pachyceras* | | | | | | | | | |
| Soil Media | 0.653 | 0.366 | 0.057 | 0.893 | 0.481 | 0.787 | 0.806 | 0.934 | 0.200 |
| Bacterial Inoculation | 0.114 | 0.058 | 0.139 | 0.001 | 0.002 | <0.001 | 0.009 | 0.010 | 0.131 |
| Soil Media * Bacterial Inoculation | 0.090 | 0.084 | 0.139 | 0.195 | 0.037 | 0.031 | 0.084 | 0.007 | 0.784 |

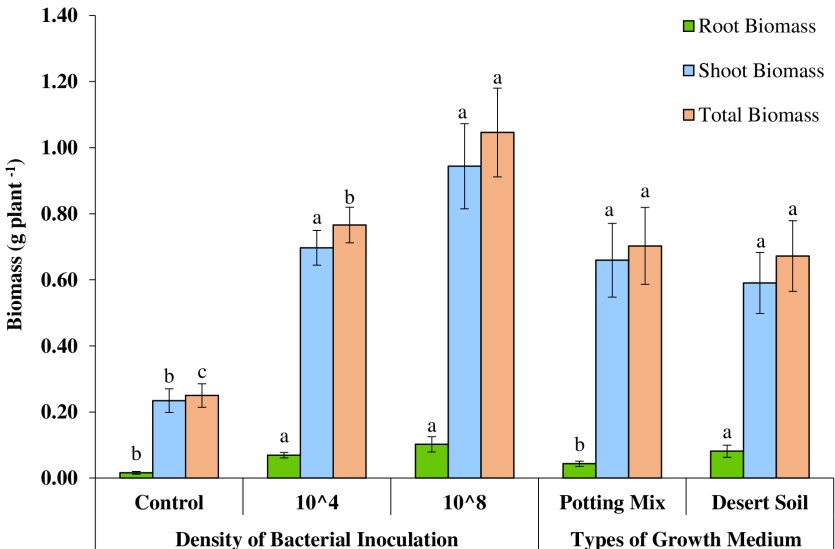

**Fig 1. Effect of type of growth medium and density of bacterial inoculation on biomass per plant in *Rhanterium epapposum*.**

Nevertheless, the average plant height, stem diameter, shoot biomass, and total biomass of *F. aegyptia* seedlings grown in desert soil increased by 26%, 24%, 58%, and 58%, respectively, compared to seedlings grown in the potting-mix medium (Fig 2). A significant increase in shoot biomass (177%) and total biomass (182%) was observed in seedlings inoculated with the indigenous bacterial inoculum at a density of $10^8$ CFU/mL cells compared to the control (Fig 2). Although inoculation with the $10^4$ CFU/mL cell density resulted in increases in these parameters, the improvements were not observed statistically significant relative to the control plants.

### H. salicornicum

Interestingly, the seedling growth substrate had a remarkable effect, significantly increased the average root biomass ($P \leq 0.007$), shoot biomass ($P \leq 0.001$), total biomass ($P < 0.001$), and root-to-shoot ratio ($P \leq 0.009$) of *H. salicornicum* seedlings (Table 1). Bacterial inoculation at both $10^4$ CFU/mL and $10^8$ CFU/mL cell densities also significantly increased shoot biomass ($P \leq 0.001$), and total biomass ($P \leq 0.001$), compared to the non-inoculated control (Table 1). However, neither growth substrate nor bacterial inoculation significantly affected the other measured growth parameters compared to the control (Table 1).

The average root biomass, shoot biomass, and total biomass of *H. salicornicum* seedlings grown in desert soil increased by 476%, 46%, and 75%, respectively, compared with seedlings grown in the potting-mix medium (Fig 3). The average shoot biomass significantly increased by 89% and 123% in seedlings inoculated with indigenous bacterial inoculum at densities of $10^4$ CFU/mL and $10^8$ CFU/mL cells, respectively (Fig 3). Similarly, the average total biomass significantly increased by 95% and 123% in seedlings inoculated with $10^4$ CFU/mL and $10^8$ CFU/mL cell densities, respectively (Fig 3). However, there was no significant difference between the two inoculum densities with respect to shoot or total biomass.

### V. pachyceras

There was a significant increase in average root biomass ($P \leq 0.001$), shoot biomass ($P \leq 0.002$), total biomass ($P < 0.001$), number of root nodules ($P \leq 0.009$), and dry weight of root nodules ($P \leq 0.010$) in seedlings inoculated with the indigenous

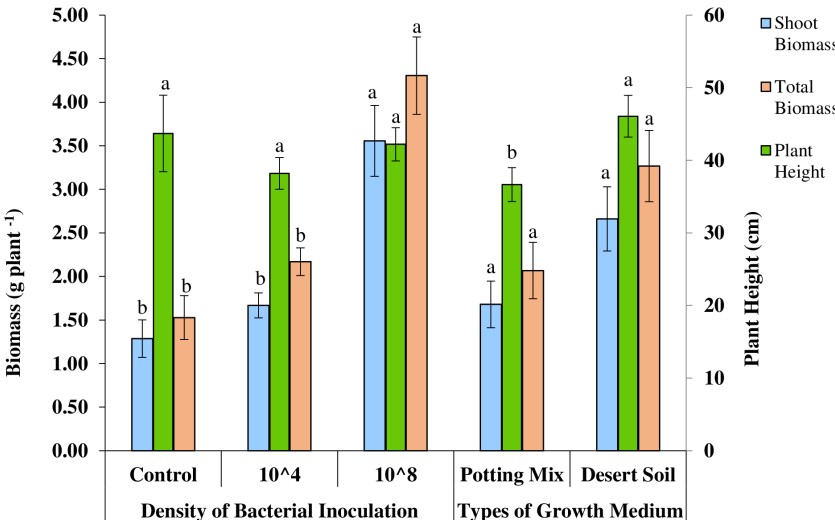

**Fig 2. Effect of type of growth medium and density of bacterial inoculation on biomass per plant, plant height and stem diameter of _Farsetia aegyptia_.**

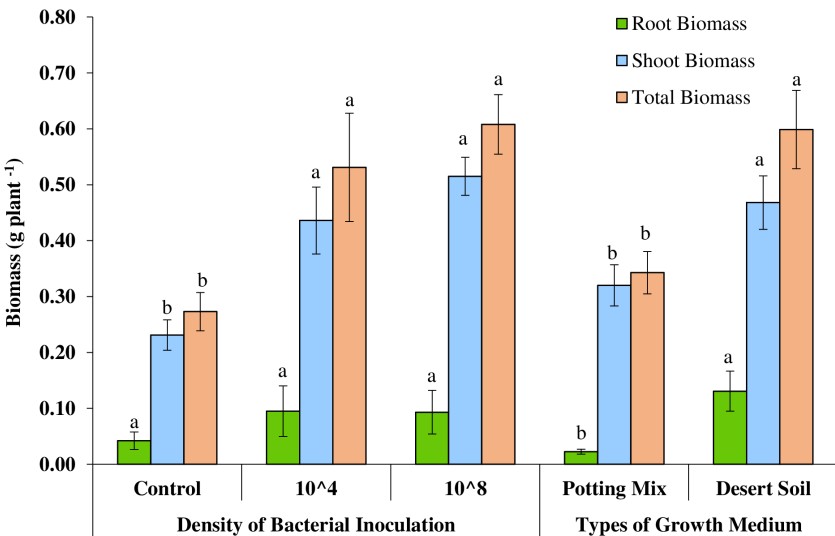

**Fig 3. Effect of type of growth medium and density of bacterial inoculation on biomass per plant of _Haloxylon salicornicum_.**

bacterial inoculum, with the $10^2$ CFU/mL and $10^4$ CFU/mL cell densities producing the most effective responses out of the four densities used, i.e., $10^2$, $10^4$, $10^6$ and $10^8$ CFU/mL cells (Table 1 and Fig 4). However, neither inoculation nor growth media had a statistically significant influence on the other recorded growth parameters (Fig 4).

The average root biomass increased by 190%, and 201% in seedlings inoculated with indigenous bacterial inoculum at densities of $10^2$ CFU/mL and $10^4$ CFU/mL cells, respectively (Fig 4). Similarly, the average shoot biomass increased by 124% and 106% the total biomass increased by 154% and 152% in seedlings inoculated with $10^2$ CFU/mL and $10^4$ CFU/mL cell densities, respectively (Fig 4). Interestingly, the highest concentration of indigenous bacterial inoculum (at $10^8$

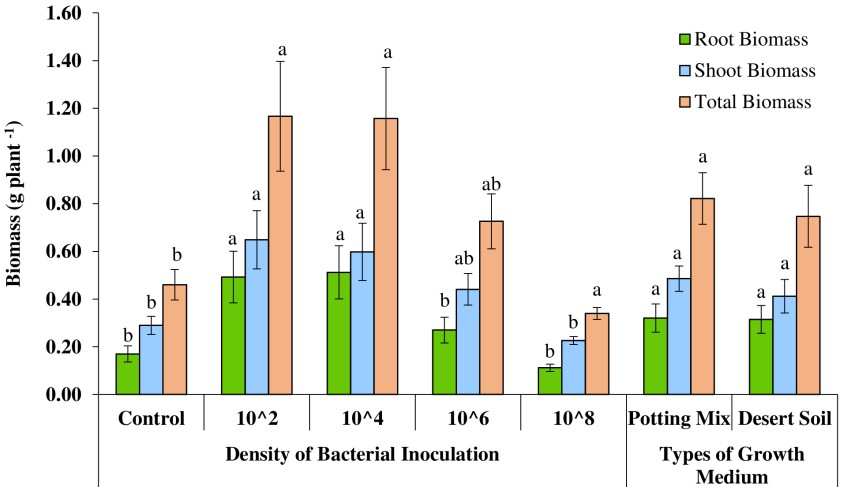

**Fig 4. Effect of type of growth medium and density of bacterial inoculation on biomass per plant of *Vachellia pachyceras*.**

CFU/mL density) had no statistically significant increase on average root biomass, shoot biomass and total biomass of the seedlings when compared to the non-inoculated control (Fig 4). The average number and dry weight of root nodules had statistically significant effect in seedlings inoculated with indigenous bacterial inoculum at densities $10^2$, $10^4$ cells.

## Effect of putative nitrogen-fixer bacterial inoculation on plant nutrient uptake

In general, inoculation of putative nitrogen-fixing bacteria at various densities significantly affected the nutrient uptake of all four selected native plant species compared to the non-inoculated control (Table 2) under greenhouse experimental conditions. However, the increase in nutrient uptake ability of the tested bacterial inoculum varied depending on the density of bacteria in the inoculum used. In contrast, there was no significant interaction effect of growing media and inoculation on the nutrient uptake except in *F. aegyptia* and *H. salicornicum* (Table 2).

### *R. epapposum*

Inoculation of *R. epapposum* with two densities ($10^4$ CFU/mL and $10^8$ CFU/mL) of putative nitrogen-fixing bacterial inoculum in the rhizosphere significantly increased shoot nitrogen (N) (P < 0.001), phosphorus (P) (P ≤ 0.001), and potassium (K) (P < 0.001) content compared to the non-inoculated control (Table 2). Specifically, shoot N content increased by 176% and 319%, shoot P content by 185% and 244%, and shoot K content by 215% and 358% when inoculated with $10^4$ CFU/mL and $10^8$ CFU/mL cell densities, respectively, using the mixture of isolated indigenous putative nitrogen-fixers (Fig 5). The cell density of $10^8$ CFU/mL produced the highest results.

Additionally, inoculation also significantly enhanced shoot magnesium (Mg) (P < 0.001), sodium (Na) (P ≤ 0.001), calcium (Ca) (P ≤ 0.001), carbon (C) (P < 0.001), and sulfur (S) (P ≤ 0.001) content (Table 2). However, neither the type of growth medium nor its interaction with bacterial inoculation had a statistically significant effect on nutrient uptake (Table 2).

### *F. aegyptia*

Inoculation of *F. aegyptia* with putative nitrogen-fixing bacterial inoculum in the rhizosphere significantly increased shoot nitrogen (N) (P < 0.001), phosphorus (P) (P < 0.001), and potassium (K) (P ≤ 0.001) uptake compared to the non-inoculated control (Table 2). Inoculation with the $10^8$ cell density resulted in statistically significant increases in shoot N, P, and K by

**Table 2. Results of Analysis of Variance (P values) Testing for the Effect Seedling Growth Medium and Bacterial Inoculation on N, P, K, Mg, Na, Ca, C and S Uptake on Selected Native Plant Species.**

| Variable | N | P | K | Mg | Na | Ca | C | S |
|---|---|---|---|---|---|---|---|---|
| *Rhanterium epapposum* | | | | | | | | |
| Soil Type | 0.573 | 0.102 | 0.342 | 0.531 | 0.160 | 0.056 | 0.717 | 0.988 |
| Bacterial Inoculation | <0.001 | 0.001 | <0.001 | <0.001 | 0.001 | 0.001 | <0.001 | <0.001 |
| Soil Type * Bacterial Inoculation | 0.532 | 0.308 | 0.700 | 0.905 | 0.538 | 0.554 | 0.367 | 0.709 |
| *Farsetia aegyptia* | | | | | | | | |
| Soil Type | <0.001 | 0.189 | 0.010 | 0.463 | 0.655 | 0.011 | 0.013 | 0.004 |
| Bacterial Inoculation | <0.001 | <0.001 | 0.001 | 0.002 | 0.105 | 0.001 | 0.001 | 0.001 |
| Soil Type * Bacterial Inoculation | 0.181 | 0.197 | 0.126 | 0.221 | 0.062 | 0.372 | 0.540 | 0.167 |
| *Haloxylon salicornicum* | | | | | | | | |
| Soil Type | 0.002 | 0.001 | 0.494 | 0.084 | 0.051 | 0.027 | 0.007 | 0.272 |
| Bacterial Inoculation | 0.014 | 0.023 | 0.002 | 0.001 | 0.007 | 0.001 | 0.004 | 0.011 |
| Soil Type * Bacterial Inoculation | 0.321 | 0.180 | 0.264 | 0.048 | 0.541 | 0.073 | 0.138 | 0.089 |
| *Vachellia pachyceras* | | | | | | | | |
| Soil Type | 0.647 | 0.804 | 0.397 | 0.275 | 0.051 | 0.308 | 0.910 | 0.313 |
| Bacterial Inoculation | 0.022 | 0.042 | 0.032 | 0.015 | 0.039 | 0.029 | 0.026 | 0.040 |
| Soil Type * Bacterial Inoculation | 0.125 | 0.186 | 0.292 | 0.506 | 0.086 | 0.229 | 0.198 | 0.131 |

N: Nitrogen; P: Phosphorus; K: Potassium; Mg: Magnesium; Na: Sodium; Ca: Calcium; C: Carbon; S: Sulphur.

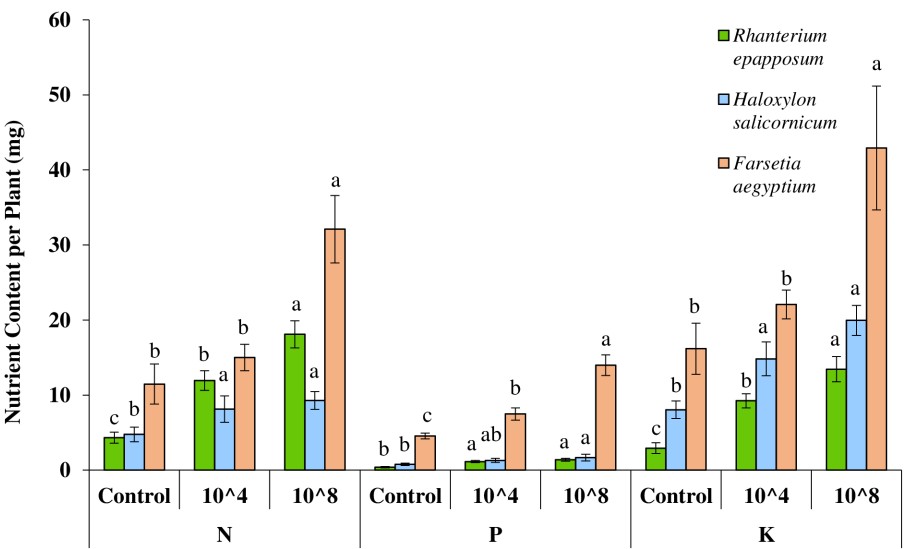

**Fig 5. Effect of bacterial inoculation on N, P, K uptake measured as shoot N, P, K content per plant in *Rhanterium epapposum, Haloxylon salicornicum* and *Farsetia aegyptia*.**

180%, 207%, and 165%, respectively (Fig 5). Furthermore, inoculation also significantly enhanced shoot uptake of magnesium (Mg) ($P \leq 0.002$), calcium (Ca) ($P \leq 0.001$), carbon (C) ($P \leq 0.001$), and sulfur (S) ($P \leq 0.001$) (Table 2).

Interestingly, the seedling growth media also had a significant effect on shoot nutrient status, influencing N ($P < 0.001$), K ($P \leq 0.010$), Ca ($P \leq 0.011$), C ($P \leq 0.013$), and S ($P \leq 0.004$) content of the plant shoot (Table 2). However, the growth media had no statistically significant effect on other parameters tested, including shoot P, Mg, and Na content (Table 2).

### *H. salicornicum*

A similar trend was observed in *H. salicornicum* as the above-mentioned native plant species. Shoot nitrogen (N) (P ≤ 0.014), phosphorus (P) (P ≤ 0.023), and potassium (K) (P ≤ 0.002) content increased significantly in seedlings inoculated with the indigenous bacterial inoculum compared to the non-inoculated control (Table 2). As shown in Fig 5, inoculation increased shoot N content by 70% and 95%, shoot P content by 68% and 115%, and shoot K content by 84% and 148% when inoculated with $10^4$ CFU/mL and $10^8$ CFU/mL cell densities, respectively. There was no statistically significant difference between the effects of $10^4$ and $10^8$ cell densities on the shoot N and K content. In addition to N, P, and K, bacterial inoculation also significantly enhanced shoot Mg (P ≤ 0.001), Na (P ≤ 0.007), Ca (P ≤ 0.001), C (P ≤ 0.004), and S (P ≤ 0.011) content (Table 2).

The growth media had a significant positive effect on several nutrient parameters. Seedlings grown in desert soil showed higher shoot N (P ≤ 0.002), Ca (P ≤ 0.027), and C (P ≤ 0.007) contents (Table 2), whereas those grown in potting soil mix recorded significantly higher shoot P (P < 0.001). In contrast, the growth media had no statically significant influence on shoot K, Mg, Na, or S contents (Table 2).

### *V. pachyceras*

Inoculation of *V. pachyceras* seedlings with different densities of indigenous bacterial inoculum resulted in statistically significant increases in shoot N (P ≤ 0.022), P (P ≤ 0.042), K (P ≤ 0.032), Mg (P ≤ 0.015), Na (P ≤ 0.039), Ca (P ≤ 0.029), C (P ≤ 0.026), and S (P ≤ 0.040) contents (Table 2). Among the inoculum densities, the $10^2$ CFU/mL level produced the greatest enhancement, increasing shoot N, P, and K contents by 152%, 129%, and 193%, respectively (Fig 6). Similar trends observed in shoot N, P, and K content and increased by 136%, 161%, and 141%, respectively when inoculated with $10^4$ CFU/mL density of bacterial inoculum (Fig 6). No significant effects of growth media or their interaction with bacterial inoculation were observed for any of the measured shoot nutrients (Table 2).

### Evaluation of commercial inoculum on *V. pachyceras*

The growth and nutrient uptake of *V. pachyceras* were also evaluated, after inoculation with the commercial ATCC bacterial strains, following the same procedure used for the indigenous bacterial inoculum experiments. The results are presented in Table 3.

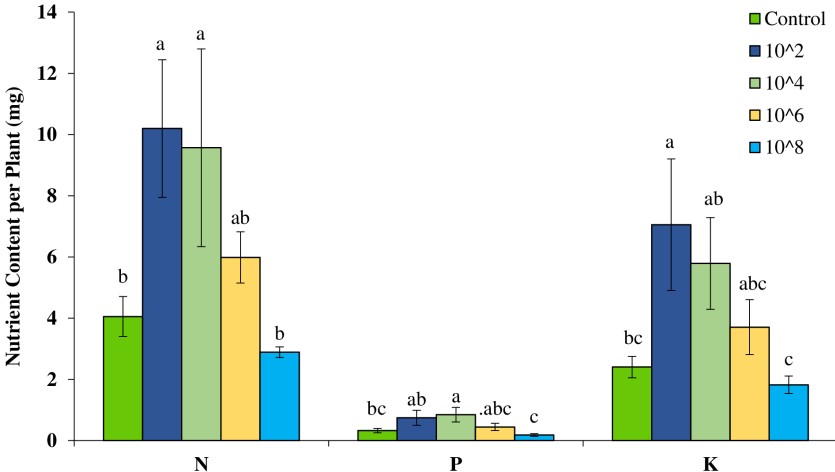

**Fig 6. Effect of bacterial inoculation on N, P, K uptake measured as shoot N, P, K content per plant in *Vachellia pachyceras*.**

**Table 3.  Results of Analysis of Variance (P values) Testing for the Effect of Commercial Inoculum on Nodulation, Growth Parameters, and Nitrogen Content of *Vachellia pachyceras*.**

| Variable | Number of Nodules | Plant Height | Shoot Biomass | N Content |
|---|---|---|---|---|
| Soil Media | 0.009 | 0.001 | 0.002 | 0.001 |
| Inoculation | 0.091 | 0.114 | 0.022 | 0.017 |
| Soil Media * Inoculation | 0.187 | 0.121 | 0.355 | 0.347 |

The seedlings grown in potting soil mix significantly increased average plant height by 75%, (P ≤ 0.001), shoot biomass by 197%, (P ≤ 0.002), number of nodules by 394% (P ≤ 0.009), and nitrogen content by 247% (P ≤ 0.001) compared to those grown in desert soil (Fig 7 and 8). The *V. pachyceras* seedlings, when inoculated with commercial *R. leguminosarum* (ATCC®10004™) and *Bradyrhizobium* sp. (ATCC®BAA-1182) significantly increased the average shoot biomass (P ≤ 0.022) and nitrogen (P ≤ 0.017) content when compared to the non-inoculated control.

The average shoot biomass of the *V. pachyceras* seedlings significantly increased by 204% and 145% when inoculated with *R. leguminosarum* ATCC strain and *Bradyrhizobium* sp. ATCC strain, respectively, when compared to the non-inoculated control (Fig 7. Likewise, the average nitrogen content of *V. pachyceras* seedlings significantly increased by 229% and 138% when inoculated with *R. leguminosarum* ATCC strain and *Bradyrhizobium* sp. ATCC strain, respectively, compared to the non-inoculated control (Fig 7).

## Discussion

Achieving natural regeneration or even establishing planted seedlings in degraded desert lands is extremely challenging, particularly in areas with highly disturbed soils, unprotected environments, and poor physicochemical properties, alongside diminishing conditions of their functional microbial communities. The current research evaluated the influence of inoculating indigenous free-living diazotrophs and root-associated plant PGPB consortia could enhance early growth performance and nutrient uptake ability of four native plant species. The indigenous bioinoculants used in this study were isolated from the rhizosphere soils associated with three native keystone plant species, along with rhizobacteria obtained

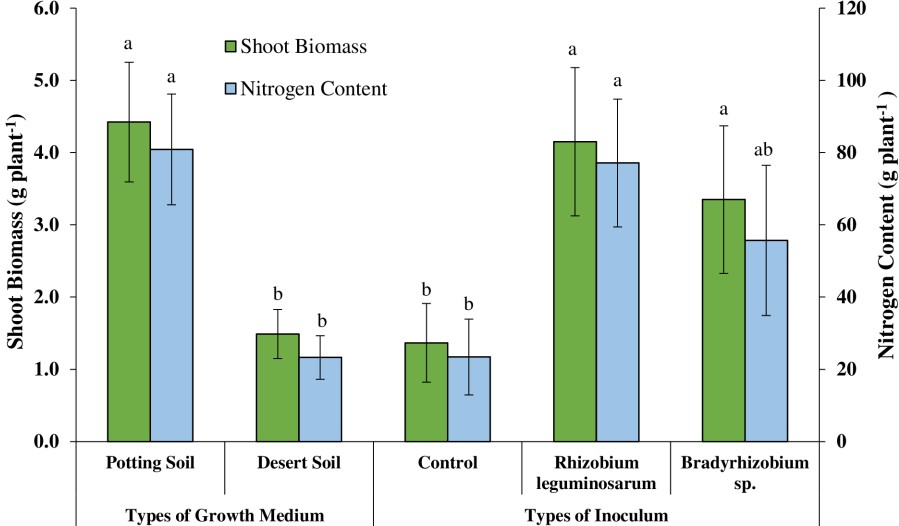

**Fig 7.  Effect of type of growth medium and commercial inoculum on shoot biomass and nitrogen content per plant of *Vachellia pachyceras*.**

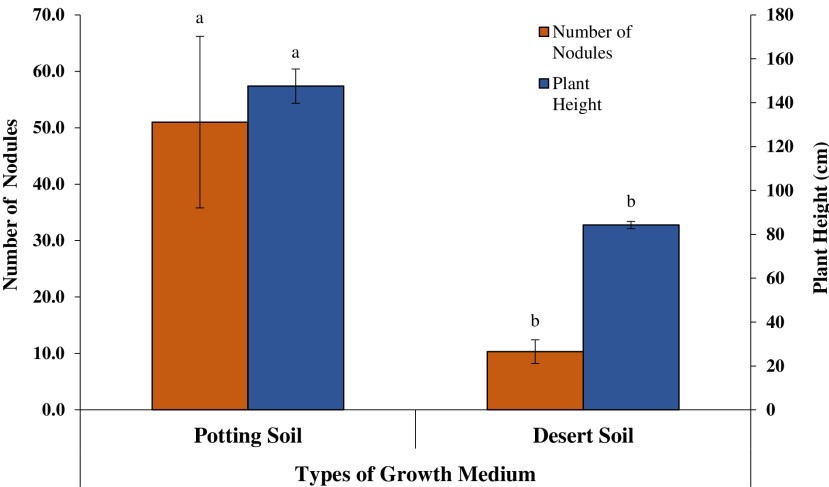

**Fig 8. Effect of type of growth medium and commercial inoculum on number of nodules and plant height of *Vachellia pachyceras*.**

from *V. pachyceras* root nodules in earlier experiment [20]. Overall, our findings show, across species, inoculation through a range of inoculum densities consistently enhanced biomass production and nutrient acquisition, indicating a broad plant growth-promoting potential of the indigenous diazotrophic consortium. These responses varied among plant species, inoculum density, and growth medium, highlighting the importance of optimizing inoculum characteristics and species selection in restoration applications. Several studies have isolated and characterized beneficial microbes, including mycorrhizal fungi and plant-growth promoting rhizibacteria from arid and semi-arid soils; most of these works have remained descriptive or focused largely on agricultural crop species [22,28–30]. However, information on the practical application of such beneficial microorganisms to enhance the growth and nutrient acquisition of native desert plants in the Middle Eastern regions remains limited [5,20]. The current study contributes to filling this gap by demonstrating the potential of locally adapted microbial inoculants to improve the early establishment of native plant species, offering a promising strategy for restoring degraded desert landscapes.

In *R. epapposum*, inoculation significantly improved biomass production at both inoculum densities ($10^4$ CFU/mL and $10^8$ CFU/mL) likely through enhanced nitrogen acquisition and possibly improved root system functioning or effective nutrient mobilization. Interestingly, the difference between the two inoculum densities was not significant for either root or shoot biomass, although total plant biomass for the $10^8$ CFU/mL treatment slightly exceeded that of the $10^4$ CFU/mL treatment. These results indicate that no additional benefits in total biomass growth beyond a functional threshold density, suggesting a saturation response. *F. aegyptia* and *H. salicornicum* exhibited, a similar pattern of enhanced growth responses particularly at the higher inoculum density ($10^8$ CFU/mL) in case of *F. aegyptia* and at both densities in case of *H. salicornicum* compared to non-inoculated controls, indicating species-specific sensitivity to bacterial load.

However, while the substantial biomass enhance indicates a pronounced growth-promoting effect of inoculation, the specific nitrogen fixation activity and nifH molecular mechanisms underlying the observed biomass enhancement were not directly measured in this study, therefore remain to be elucidated in future work. Nevertheless, in general, these results demonstrate that native desert plant species in Kuwait are highly responsive to inoculation with indigenous bacterial consortia. The observed increases in biomass across multiple species emphasize the potential of regionally adapted microbial inoculants to improve early seedling performance and promote growth under nutrient-limited desert conditions.

*V. pachyceras* was examined across four inoculum densities ($10^2$, $10^4$, $10^6$, and $10^8$ CFU/mL). Interestingly, V. *pachyceras* responded optimally to low to medium densities ($10^2$ CFU/mL and $10^4$ CFU/mL), exhibited significant growth performance,

as well as enhanced nodulation compared to non-inoculated controls. In contrast, the highest inoculum density ($10^8$ CFU/mL) did not enhance growth or nodulation. This pattern suggests density-dependent regulation of symbiosis, potentially linked to a possible saturation effect, carbon costs, microbial competition, or oxygen limitations affecting symbiotic efficiency. Biological nitrogen fixation is an energetically expensive process in which atmospheric $N_2$ is converted into ammonia utilizing ATP, which requires carbon investment from the host plant [31]. The high inoculum densities may impose a carbon burden that outweighs the benefits of additional bacterial cells. This may explain reduced growth response in *V. pachyceras* at $10^8$ CFU/mL treatment level.

These findings are consistent with previous observations reporting density-dependent outcomes in legume-rhizobia symbioses [15,28,32–34], reinforcing the significance of plants optimizing inoculum density to achieve maximum plant benefit. In the context of restoration and revegetation programs, the results indicate that moderate inoculum densities are sufficient to enhance seedling establishment while maintaining cost-effectiveness.

In our experiment, the growth medium significantly influenced plant performance, particularly root development across the evaluated species, with pronounced effects in *R. epapposum* and *H. salicornicum*. Root biomass increased by 88% in *R. epapposum* and by 476% in *H. salicornicum* when grown in desert soils compared to potting mix soils, indicating species-specific substrate responses. Enhanced root development is particularly advantageous in arid environments, where deeper or more extensive root systems improve water and nutrient acquisition under limiting conditions. These findings suggest a strong species-dependent response, and also highlights soil physiochemical properties, including soil structure, nutrient availability, species native ecological growing conditions, and native microbial composition, which may substantially influence root growth, proliferation, and plant responsiveness to microbial inoculation. The superior performance of *H. salicornicum* and *F. aegyptia* in desert soil likely reflects ecological adaptation to native edaphic conditions and greater compatiblility with indigenous microbial communities. However, the effectiveness of microbial inoculation ultimately depends on successful root colonization, which is shaped by soil environment, particular ecological niche and the degree of co-adaptation between host and microbial partners [35,36].

Our results suggest that inoculation success depends on a conducive soil environment that supports effective roots colonization and symbiosis. Pankievicz *et al*. [31] indicated that efficient symbiosis requires compatibility between host plants and diazotrophic bacteria under favorable environmental conditions that support optimal nitrogen fixation. The positive response to indigenous rhizospheric inoculum further suggests that native bacterial isolates are well adapted to arid, nutrient-poor soils and remain functionally effective under greenhouse conditions.

Across all species, bacterial inoculation significantly increased plant nutrient content, particularly N, P, K, and key micronutrients (Mg, Ca, Na, and S). These increases likely reflect both direct microbial contributions to nutrient supply and indirect effects, including improved root development and rhizosphere nutrient mobilization. Our findings align with previous findings reporting that PGPR enhance growth and nutrient uptake of maize plants when inoculated under greenhouse conditions [37,38], and improved plant establishment of native plants and soil quality in degraded forest and desert soils [15–17]. Similarly, a recent report demonstrated that nitrogen-fixing bacteria isolated from giant reed, and switch grass have the potential to influence plant growth and total nutrient uptake when inoculated into agricultural crops [39]. PGPR-based inoculants and biostumulants have also been reported to improve wheat growth [40], increase crop yield and nutrient uptake [41], and enhance N, P, and K uptake in maize grown in nutrient-deficient calcisol soil [28]. Tsegaye et al. [29] further demonstrated that both single-strained consortium PGPR treatments significantly improved teff growth, grain yield, and uptake of N, P, K, Ca, and S. Consistent with the findings of Egamberdiyeva [28] and Xu *et al*. [39], our findings confirm that bacterial inoculation enhances not only macronutrient uptake but also the accumulation of micronutrients Mg, Na, Ca, C, and trace elements.

In *R. epapposum*, inoculation significantly increased N (176–319%), P (185–244%), and K (215–358%) uptake, with the highest inoculum density yielding the strongest response. Nutrient uptake was unaffected by growth medium, or interaction with inoculation (Table 2), indicating consistent bacterial effect across the soil types. Similarly, *F. aegyptia,* exhibited

inoculation-induced increases in nutrient uptake as observed for increased biomass production. However, growth medium significantly influenced N, K, Ca, and S uptake, suggesting that the soil's inherent nutrient or physiochemical properties and compatibility with indigenous microbial communities influenced the plant-microbe interactions. The superior performance at $10^8$ CFU/mL treatment implies that a sufficiently high bacterial load may be necessary for maximizing benefits in this species. These results suggest that optimizing inoculum density and matching soil conditions are critical for maximizing benefits in *F. aegyptia.*

In *H. salicornicum,* inoculation significantly enhanced N (70–95%), P (68–115%), and K (84–148%) uptake, along with improved micronutrient assimilation, particularly at higher inoculum density. These results suggest that inoculation may facilitate mobilization of multiple macro and micronutrients in nutrient-poor desert soils. The greater response at $10^8$ CFU/mL versus $10^4$ CFU/mL treatment suggests that higher inoculum density is beneficial in this species. In contrast, *V. pachyceras,* showed maximum nutrient uptake at lower inoculum density ($10^2$–$10^4$ CFU/mL). While inoculation significantly increased shoot N, P, K, Mg, Na, Ca, C, and S, the highest density ($10^8$ CFU/mL) did not further enhance N. P, or K uptake, indicating possible density-dependent regulation or competitive effects. The responses in *Vachellia* system indicate a more complex symbiotic system in which isolated root-nodulating bacteria establish effective associations with the host plant primarily at low to moderate densities, likely achieving optimal root colonization and balanced plant-microbe interactions. Such responses suggest that these isolates possess full or partial symbiotic capabilities either forming functional nodules and contributing directly to biological nitrogen fixation or promoting plant growth via indirect mechanisms such as phytohormore production, nutrient solubilization, or root architecture modifications. Both indigenous isolates and commercial rhizobial strains significantly enhanced growth and nutrient content under greenhouse conditions, confirming effective symbiosis and N-fixation potential.

Inoculation with commercial strains of *R. leguminosarum* ATCC and *Bradyrhizobium* sp. ATCC increased shoot biomass and nitrogen content compared to non-inoculated controls, with *R. leguminosarum* ATCC showing the strongest response (Table 3), indicating effective symbioses and high N-fixation potential. Growth medium also influenced plant performance, as seedlings grown in potting soil exhibited greater height, shoot biomass, nodule number, and nitrogen content than those in desert soil, likely due to more favorable physicochemical conditions. Although greenhouse results demonstrate the potential promise of these commercial inoculants, field validation under arid-land conditions is necessary. Overall, this greenhouse inoculation study demonstrates that inoculation with nitrogen-fixing bacteria can substantially enhance the growth and nutrient acquisition of desert native plants in nutrient-poor soils and nursery substrates. Consistent with previous studies, these improvements likely reflect integrated mechanisms including biological N-fixation, improved root development, increased root exudation, nutrient solubilization, and rhizosphere activation, and phytohormone production [42–45]. Although the specific mechanisms operating in this study cannot be conclusively identified within the scope of this study, previous studies attribute similar improvements to plant growth promotion, biological N fixation, organic compounds production, disease suppression, and root elongation [28,46]. Other studies have reported that PGPB can suppress soil-borne pathogens and improve soil structure and microbial diversity, thereby indirectly supporting nutrient absorption and seedling establishment [18,19]. Increased total nutrient uptake may also reflect increased shoot and root biomass as observed in agricultural crops [47,48]. To our knowledge, this is the first study assessing growth response and nutrient mobilization of *V. pachyceras* inoculated with nitrogen-fixing rhizobacteria isolated from the Kuwait desert, highlighting their potential application as biofertilizers for native plants establishment.

## Conclusion

In general, indigenous diazotrophic and commercial rhizobial inoculants improved early seedling growth and nutrient uptake of Kuwait desert species under the current experimental conditions. The study evaluated the collective effects of isolated N-fixing bacteria rather than individual strains, demonstrating that growth improvements, successful nodulation, and N-fixation depend on host-microbe compatibility, inoculum density, and species-specific responses. The inoculation

protocol was effective across all tested native species, highlighting the potential of these isolates to enhance growth and nutrient acquisition in Kuwait's desert flora and to support effective revegetation of degraded arid lands. Moderate inoculum densities were often sufficient to achieve positive responses, suggesting practical relevance for restoration initiatives. Despite the promising greenhouse findings, several limitations should be acknowledged. The controlled greenhouse conditions may not fully represent field environments. Soil heterogeneity, resident microbial competition, and environmental stresses can influence inoculant performance. Therefore, field-based validations are required to determine long-term effectiveness of indigenous and commercial inoculants under arid environmental conditions. Such research will support the development of optimized, field-ready bioinoculation strategies for sustainable restoration and revegetation efforts in desert ecosystems.

## Supporting information

**S1 Table. Indigenous Nitrogen Fixing Bacterial Isolates Used for the Preparation of Inoculum.**
(DOCX)

**S2 Table. Physio-Chemical Properties of Desert Soil Collected from KISR's Station for Research and Innovation (KSRI).**
(DOCX)

## Acknowledgments

The authors immensely acknowledge the Kuwait Institute for Scientific Research (KISR) and Kuwait Foundation for the Advancement of Sciences (KFAS) for the constant support and encouragement throughout the project. We also thank the greenhouse helpers for their assistance in the maintenance of greenhouse experiments and laboratory helpers for laboratory analysis.

## Author contributions

**Conceptualization:** Majda K. Suleiman, Ali M. Quoreshi.

**Formal analysis:** Anitha J. Manuvel, Mini T. Sivadasan, Sheena Jacob.

**Funding acquisition:** Majda K. Suleiman, Ali M. Quoreshi.

**Investigation:** Majda K. Suleiman, Ali M. Quoreshi, Anitha J. Manuvel, Mini T. Sivadasan, Sheena Jacob.

**Methodology:** Majda K. Suleiman, Ali M. Quoreshi, Anitha J. Manuvel, Mini T. Sivadasan.

**Project administration:** Majda K. Suleiman.

**Supervision:** Majda K. Suleiman, Ali M. Quoreshi.

**Validation:** Majda K. Suleiman, Ali M. Quoreshi.

**Writing – original draft:** Majda K. Suleiman, Ali M. Quoreshi, Anitha J. Manuvel.

**Writing – review & editing:** Majda K. Suleiman, Ali M. Quoreshi, Anitha J. Manuvel, Mini T. Sivadasan, Sheena Jacob.

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
