## [Decision Letter · Decision Letter 0]

21 Jan 2026

PONE-D-25-64029Inoculation of indigenous nitrogen-fixers isolated from the Kuwait desert enhances seedling growth and nutrient uptake in a greenhouse bioassayPLOS One

Dear Dr. Quoreshi,

Thank you for submitting your manuscript to PLOS ONE. After careful consideration, we feel that it has merit but does not fully meet PLOS ONE’s publication criteria as it currently stands. Therefore, we invite you to submit a revised version of the manuscript that addresses the points raised during the review process.

We look forward to receiving your revised manuscript.

Kind regards,

Vishal Tripathi, Ph.D.

Academic Editor

PLOS One

“This study was funded by Kuwait Foundation for the Advancement of Sciences (KFAS). The grand number is Project – P215-42SL-01”

Reviewers' comments:

Reviewer's Responses to Questions

**Comments to the Author**

1. Is the manuscript technically sound, and do the data support the conclusions?

Reviewer #1: Yes

Reviewer #2: Yes

2. Has the statistical analysis been performed appropriately and rigorously? 

Reviewer #1: No

Reviewer #2: Yes

3. Have the authors made all data underlying the findings in their manuscript fully available?

The PLOS Data policy requires authors to make all data underlying the findings described in their manuscript fully available without restriction, with rare exception (please refer to the Data Availability Statement in the manuscript PDF file). The data should be provided as part of the manuscript or its supporting information, or deposited to a public repository. For example, in addition to summary statistics, the data points behind means, medians and variance measures should be available. If there are restrictions on publicly sharing data—e.g. participant privacy or use of data from a third party—those must be specified.requires authors to make all data underlying the findings described in their manuscript fully available without restriction, with rare exception (please refer to the Data Availability Statement in the manuscript PDF file). The data should be provided as part of the manuscript or its supporting information, or deposited to a public repository. For example, in addition to summary statistics, the data points behind means, medians and variance measures should be available. If there are restrictions on publicly sharing data—e.g. participant privacy or use of data from a third party—those must be specified.requires authors to make all data underlying the findings described in their manuscript fully available without restriction, with rare exception (please refer to the Data Availability Statement in the manuscript PDF file). The data should be provided as part of the manuscript or its supporting information, or deposited to a public repository. For example, in addition to summary statistics, the data points behind means, medians and variance measures should be available. If there are restrictions on publicly sharing data—e.g. participant privacy or use of data from a third party—those must be specified.requires authors to make all data underlying the findings described in their manuscript fully available without restriction, with rare exception (please refer to the Data Availability Statement in the manuscript PDF file). The data should be provided as part of the manuscript or its supporting information, or deposited to a public repository. For example, in addition to summary statistics, the data points behind means, medians and variance measures should be available. If there are restrictions on publicly sharing data—e.g. participant privacy or use of data from a third party—those must be specified.

Reviewer #1: Yes

Reviewer #2: Yes

4. Is the manuscript presented in an intelligible fashion and written in standard English?

Reviewer #1: No

Reviewer #2: No

5. Review Comments to the Author

Reviewer #1: The manuscript and data presented is quite interesting but the writing or presentation is wanting. I have left my review comments in the attached manuscript file. Most glaring issues are as follows: The standard rules of writing scientific names are not followed in the document. The introduction section is not clear but contains information that is jumbled up with no clear flow from the general background, to the problem at hand, to the research gap and the aim of the study. The results presented in the tables should indicate the means plus or minus the standard deviations assuming the measurements were conducted in triplicates as should be the case. The labels in the figures are not legible. Figures 7 and 8 are a bit chaotic because they contain different variables with different measurement units but are graphed together e.g plant height and number of nodules in Figure 8. This shouldn't be the case at all. The rest of the comments are in the attachemnt

Reviewer #2: This manuscript provides valuable and novel insights into the use of indigenous diazotrophic bacteria to enhance the growth and nutrient uptake of native plant species in arid ecosystems. The topic aligns well with restoration ecology priorities and is technically sound.

Major Strengths:

- Novel and relevant contribution to arid-land restoration

- Strong experimental design with factorial approach

- Clear evidence of treatment effects supported by data

- Practical implications for restoration and biofertilizer applications

Major Points for Improvement:

1. Mechanistic interpretations are inferred. Authors should clarify limitations and suggest future validation.

2. Statistical assumptions (normality/variance) are not reported. It is recommended to include.

3. Desert soil physicochemical properties are not provided. It is suggested that a soil properties table be added.

4. Composition of inoculum mixtures not fully detailed. It is suggested to include isolate IDs and genera.

Minor Comments:

- Standardize formatting of CFU units, superscripts, and species italics

- Improve clarity of figure axis labels

- Minor grammatical smoothing recommended

More specfic section wise comments are as follows:

TITLE / ABSTRACT

Page 1, line 1–2:

The title uses “enhances seedling growth and nutrient uptake in a greenhouse bioassay. “Consider adding 'indigenous nitrogen-fixers' earlier for indexing clarity, for example: 'Inoculation with indigenous nitrogen-fixers enhances seedling growth…”

Page 2, lines 24–28:

Abstract introduces desert soil degradation but repeats phrases such as “nutrient-poor soil systems”. Tighten to improve flow by merging lines 24–29 into a more concise problem definition.

Page 2, lines 39–40:

Typo in “thereby supporting their potential use them as biofertilizer…” Remove “them” “…supporting their potential use as biofertilizers…”

INTRODUCTION

Page 3, lines 52–63:

Very descriptive and slightly repetitive regarding desertification and native vegetation decline. Condense by removing repeated causes (climate, anthropogenic effects) already mentioned above.

Page 3, line 61–62:

“soil microbial community [1]”. Citation [1] does not appear to support soil microbial loss specifically. Ensure citations align with the stated claim or modify wording.

Page 4, lines 82–85:

Diazotroph definition could be streamlined; sentences are long. This can be revised, for example: “Diazotrophs are rhizospheric or root-associated bacteria capable of fixing atmospheric nitrogen into plant-available forms.”

Page 5, lines 102–103:

“obtained from our previous study [20]” is useful, but no brief summary of identification outcomes. Add one sentence explaining how many strains and which genera were isolated to improve continuity.

MATERIALS & METHODS

Page 6, lines 117–122:

Comment: Cell concentrations and pooling are described, but the genera/species identities of isolates are not listed. Add a supplemental table summarizing: strain code, genus, isolation source, and 16S accession numbers (if applicable).

Page 7, lines 132–138:

Soil preparation is described, but no soil physicochemical properties are provided. Add a soil property table (pH, EC, OM%, total N, P, texture), as these directly affect the interpretation of nutrient uptake.

Page 7, lines 144–151:

Seed sterilization and scarification methods are detailed but lack justification for differences across species. Add at least a sentence explaining why sulfuric acid treatment was used only for Vachellia pachyceras.

Page 8, lines 160–167:

The commercial inoculum strains are listed only as ATCC numbers. Also provide scientific names, e.g., Rhizobium leguminosarum bv. viciae…

Page 8, lines 171–182:

Comment: Statistical methods are adequate, but missing assumption verification details. Add information on the following tests:

• Shapiro–Wilk or KS test for normality

• Levene’s test for homogeneity of variances

• Effect size metrics (optional but recommended)

RESULTS

Page 10–15, Tables 1 & 2:

Clear presentation, but units for biomass (e.g., g plant⁻¹) are not specified. Add units beneath column headers for clarity.

Page 12, line 214–217:

“did not have any significant effect…” appears multiple times across species. Consider reporting non-significant trends if biologically relevant.

Figures (p.11–18):

Figures lack clear axis labels (units for biomass, nutrient concentration). Add units (e.g., mg plant⁻¹ for nutrients, g for biomass).

DISCUSSION

Page 19–21:

Very detailed, but occasionally repetitive on “biomass increased significantly…” Combine repetitive biomass comparisons and focus more on ecological interpretation.

Page 21, lines 397–399:

Growth responses are linked to nitrogen acquisition, but no mechanisms were measured. Add a sentence acknowledging mechanistic limitations (e.g., no ARA post-inoculation, nifH expression).

Page 22, lines 418–423:

“Biological nitrogen fixation is an energetically expensive process…”. It is a good context. However, add a citation to support the discussion of metabolic cost.

Page 24–26:

Discussion of nutrient uptake improvements is strong, but does not distinguish direct vs. indirect mechanisms. Add a 1–2 sentence hypothesis section:

• Direct: N fixation, P solubilization

• Indirect: phytohormones, root architecture changes

CONCLUSION

Page 27, lines 535–546:

Strong applied relevance but lacks explicit limitations to improve realism and strengthen the case for future field trials. The following limitations can be added:

• Greenhouse vs. field performance

• Microbe persistence in natural soils

• Competition with native microbiota

REFERENCE SECTION

Page 28 onward:

Ensure consistency in formatting (journal names sometimes abbreviated, sometimes not). Adapt to PLOS ONE reference formatting guidelines.

6. PLOS authors have the option to publish the peer review history of their article (what does this mean?). If published, this will include your full peer review and any attached files.). If published, this will include your full peer review and any attached files.). If published, this will include your full peer review and any attached files.). If published, this will include your full peer review and any attached files.

...

Reviewer #1: **Yes:** Dr. Becky AlooDr. Becky AlooDr. Becky AlooDr. Becky Aloo

Reviewer #2: No

---

## [Author Response · Author response to Decision Letter 1]

5 Mar 2026

Inoculation of indigenous nitrogen-fixers isolated from the Kuwait desert enhances seedling growth and nutrient uptake in a greenhouse bioassay

Revised Title: “Inoculation with indigenous nitrogen-fixers enhances seedling growth and nutrient uptake in a greenhouse bioassay”

First Revision comments

General Comments (from e-mail)

Comment 1. Please ensure that your manuscript meets PLOS ONE's style requirements, including those for file naming. The PLOS ONE style templates can be found at

Response 1: Thank you for your suggestion. The revised manuscript has been carefully reviewed and now complies with all PLOS ONE’s style requirements outlined in the links provided above.

Comment 2. In your Methods section, please provide additional information regarding the permits you obtained for the work. Please ensure you have included the full name of the authority that approved the field site access and, if no permits were required, a brief statement explaining why.

Response 2: The desert soil used as growth medium in this greenhouse assay was collected from the KISR’s Station for Research and Innovation (KSRI). No permit was required for site access sample collection, as the research station is owned and operated by Kuwait Institute for Scientific Research (KISR).

Comment 3. Thank you for stating the following financial disclosure:

“This study was funded by Kuwait Foundation for the Advancement of Sciences (KFAS). The grand number is Project – P215-42SL-01”

Response 3: As mentioned, the following statement was added in the revised cover letter and uploaded.

Comment 4. If the reviewer comments include a recommendation to cite specific previously published works, please review and evaluate these publications to determine whether they are relevant and should be cited. There is no requirement to cite these works unless the editor has indicated otherwise.

Response 4: Thanks for the comment. We carefully reviewed concern citation. The current citations in the revised manuscript are correct and relevant.

Comment 5. Review Comments to the Author

Reviewer #1: The manuscript and data presented is quite interesting but the writing or presentation is wanting. I have left my review comments in the attached manuscript file. Most glaring issues are as follows: The standard rules of writing scientific names are not followed in the document. The introduction section is not clear but contains information that is jumbled up with no clear flow from the general background, to the problem at hand, to the research gap and the aim of the study. The results presented in the tables should indicate the means plus or minus the standard deviations assuming the measurements were conducted in triplicates as should be the case. The labels in the figures are not legible. Figures 7 and 8 are a bit chaotic because they contain different variables with different measurement units but are graphed together e.g plant height and number of nodules in Figure 8. This shouldn't be the case at all. The rest of the comments are in the attachment.

Response 5 (a): We sincerely thank the reviewer for the constructive and insightful comments. All remarks and the reviewer comments in the annotated manuscript have been carefully addressed in the revised version. Specifically, scientific names have been thoroughly reviewed and corrected throughout the manuscript to ensure compliance with standard nomenclature conventions. The introduction section has been substantially reorganized, revised thoroughly, removed repetitions to improve clarity and logical flow, clearly establish background to research problems, identified knowledge gaps, and objective of the study.

Regarding the results section, the data provided in the tables are p values derived from statistical analyses. Therefore, the values presented in these tables should not present as mean ± standard deviation as well as presenting units are not relevant. Additional statistical analysis was performed as commented and all the figures have been improved and presented in a clearer and more appropriate format in the revised manuscript.

We believe these revisions have significantly strengthened the manuscript.

Reviewer #2: This manuscript provides valuable and novel insights into the use of indigenous diazotrophic bacteria to enhance the growth and nutrient uptake of native plant species in arid ecosystems. The topic aligns well with restoration ecology priorities and is technically sound.

Major Strengths:

- Novel and relevant contribution to arid-land restoration

- Strong experimental design with factorial approach

- Clear evidence of treatment effects supported by data

- Practical implications for restoration and biofertilizer applications

Response (5b):

We sincerely thank the reviewer for the positive evaluation of our manuscript and for recognizing its novelty, methodological strength, and practical relevance. We greatly appreciate the acknowledgment of its contribution to arid-land restoration and biofertilizer applications. Your encouraging feedback is highly valued.

Major Points for Improvement:

1. Mechanistic interpretations are inferred. Authors should clarify limitations and suggest future validation.

Response: The result section was carefully revised to avoid any mechanistic interpretations and presented in the revised manuscript

2. Statistical assumptions (normality/variance) are not reported. It is recommended to include.

Response: Prior to analysis, data normality and homogeneity was verified. As recommended this information is now reported in the Data Collection and Data Analysis section.

3. Desert soil physicochemical properties are not provided. It is suggested that a soil properties table be added.

Response: Thank you for this valuable suggestion. The desert soil used in this study was collected from the KSRI. As recommended, a table summarizing the soil physiochemical properties of the soil has now been added to the Supplementary Materials Section (Table S2) in the revised manuscript.

4. Composition of inoculum mixtures not fully detailed. It is suggested to include isolate IDs and genera.

Response: Thanks for the helpful comment. The isolate IDs corresponding genera used in this experiment were previously published in our earlier study and are cited in the inoculum preparation section of the manuscript. To improve clarity and completeness, we have now added a supplementary table detailing the composition of the inoculum mixture including isolate ID and respective genera in the Supplementary Materials Section (Table S1).

Minor Comments:

- Standardize formatting of CFU units, superscripts, and species italics

Response: We apologize for the oversight. All CFU units, superscripts, and scientific names have been carefully reviewed and corrected to ensure consistence and adherence to standard formatting conventions throughout the revised manuscript.

- Improve clarity of figure axis labels

Response: The figure axis labels have been revised and improved to enhance the clarity and readability in the revised manuscript.

- Minor grammatical smoothing recommended

Response: As recommended, the manuscript has been thoroughly proofread and minor grammatical issues have been corrected to improve overall clarity and flow.

More specific section wise comments are as follows: (From e-mail)

Line number mentioned below are reflected as per original submission of the manuscript.

TITLE / ABSTRACT

Comment 6. Page 1, line 1–2:

The title uses “enhances seedling growth and nutrient uptake in a greenhouse bioassay. “Consider adding 'indigenous nitrogen-fixers' earlier for indexing clarity, for example: 'Inoculation with indigenous nitrogen-fixers enhances seedling growth…”

Response 6: We appreciate the suggestion on modifying the title for indexing clarity. As suggested the title has been revised to improve indexing clarity and now reads: “Inoculation with indigenous nitrogen-fixers enhances seedling growth and nutrient uptake in a greenhouse bioassay” in the revised manuscript.

Comment 7. Page 2, lines 24–28:

Abstract introduces desert soil degradation but repeats phrases such as “nutrient-poor soil systems”. Tighten to improve flow by merging lines 24–29 into a more concise problem definition.

Response 7: Thank you for this helpful comment. We have revised the overall abstract, particularly line 24-29 were re-written to improve clarity and flow by eliminating repetitive phrasing and merging the indicated lines into a more concise problem statement in the revised manuscript.

Comment 8. Page 2, lines 39–40:

Typo in “thereby supporting their potential use them as biofertilizer…” Remove “them” “…supporting their potential use as biofertilizers…”

Response 8: Corrected as suggested. The sentence now read “…. supporting their potential use as biofertilizer.”

INTRODUCTION

Comment 9. Page 3, lines 52–63:

Very descriptive and slightly repetitive regarding desertification and native vegetation decline. Condense by removing repeated causes (climate, anthropogenic effects) already mentioned above.

Response 9: Thank you for your suggestions. Introduction section has been thoroughly revised, condensed to remove repetition of previously states drivers of desertification throughout the introduction section, while keeping the key contextual information.

Comment 10. Page 3, line 61–62:

“soil microbial community [1]”. Citation [1] does not appear to support soil microbial loss specifically. Ensure citations align with the stated claim or modify wording.

Response 10: We agree and revised the sentence to better reflect the cited reference, where appropriated.

Comment 11. Page 4, lines 82–85:

Diazotroph definition could be streamlined; sentences are long. This can be revised, for example: “Diazotrophs are rhizospheric or root-associated bacteria capable of fixing atmospheric nitrogen into plant-available forms.”

Response 11: As suggested, the definition of diazotroph has been rewritten for clarity and conciseness and incorporated in the revised manuscript.

Comment 12. Page 5, lines 102–103:

“obtained from our previous study [20]” is useful, but no brief summary of identification outcomes. Add one sentence explaining how many strains and which genera were isolated to improve continuity.

Response 12: We understand the reviewer’s concern regarding the identification of bacterial strains used in this study as well as to improve the continuity with the current study. To address this comment a table summarizing the details of bacterial isolates and their major genera used in this experiment has been added in the supplementary section (Table S1).

MATERIALS & METHODS

Comment 13. Page 6, lines 117–122:

Comment: Cell concentrations and pooling are described, but the genera/species identities of isolates are not listed. Add a supplemental table summarizing: strain code, genus, isolation source, and 16S accession numbers (if applicable).

Response 13: As suggested, a supplementary table (Table S1) summarizing: strain code, genus, isolation source, and 16S accession numbers has been added in the revised manuscript.

Comment 14. Page 7, lines 132–138:

Soil preparation is described, but no soil physicochemical properties are provided. Add a soil property table (pH, EC, OM%, total N, P, texture), as these directly affect the interpretation of nutrient uptake.

Response 14: As suggested, a new soil properties table has included in the revised manuscript as a supplementary table (Table S2) reporting the physiochemical properties of soil used for the experiment to support interpretation of nutrient uptake results.

Comment 15. Page 7, lines 144–151:

Seed sterilization and scarification methods are detailed but lack justification for differences across species. Add at least a sentence explaining why sulfuric acid treatment was used only for Vachellia pachyceras.

Response 15: As suggested, a justification sentence has been added in the revised manuscript explaining why sulfuric acid scarification was applied only to Vachellia pachyceras due to its hard, impermeable seed coat and dormancy requirement.

Comment 16. Page 8, lines 160–167:

The commercial inoculum strains are listed only as ATCC numbers. Also provide scientific names, e.g., Rhizobium leguminosarum bv. viciae…

Response16: Scientific names corresponding to the ATCC strains have added as per the brochure supplied along with the commercial bacterium, the commercial bacterial strains are cited as Rhizobium leguminosarum (ATCC®10004TM) and Bradyrhizobium sp. (ATCC®BAA-1182) in the revised manuscript.

Comment 17. Page 8, lines 171–182:

Comment: Statistical methods are adequate, but missing assumption verification details. Add information on the following tests:

• Shapiro–Wilk or KS test for normality

• Levene’s test for homogeneity of variances

• Effect size metrics (optional but recommended)

Response 17:

Thanks for your suggestion. A sentence addressing your concern has been added as” Prior to analysis, data normality was verified using Shapiro-Wilk test and homogeneity of variances using Levene’s Test” under Data Collection and Data Analysis Section in the revised manuscript.

RESULTS

Comment 18. Page 10–15, Tables 1 & 2:

Clear presentation, but units for biomass (e.g., g plant⁻¹) are not specified. Add units beneath column headers for clarity.

Response 18: Regarding the comment on the Tables 1 and 2, the data provided in the tables are p values derived from statistical analyses only and not the actual biomass data. Therefore, the values presented in these tables (p values) should not present as mean ± standard deviation and also presenting units are not relevant.

Comment 19. Page 12, line 214–217:

“did not have any significant effect…” appears multiple times across species. Consider reporting non-significant trends if biologically relevant.

Response 19: The result section has been revised to report non-significant trends wherever necessary and relevant.

Comment 20. Figures (p.11–18):

Figures lack clear axis labels (units for biomass, nutrient concentration). Add units (e.g., mg plant⁻¹ for nutrients, g for biomass).

Response 20: As suggested, we carefully reviewed all figures for its axis accuracy, axis labels. The figures axis labels are correct reflecting appropriate measurement units. We corrected axis label in Figure 7 as suggested.

DISCUSSION

Comment 21. Page 19–21:

Very detailed, but occasionally repetitive on “biomass increased significantly…” Combine repetitive biomass comparisons and focus more on ecological interpretation.

Response 21: As suggested, the discussion section has been thoroughly reviewed and revised to reduce repetitive reporting of biomass increases and strengthen ecological interpretation of the findings.

Comment 22. Page 21, lines 397–399:

Growth responses are linked to nitrogen acquisition, but no mechanisms were measured. Add a sentence acknowledging mechanistic limitations (e.g., no ARA post-inoculation, nifH expression).

Response 22: We appreciate your comments and agreed mechanis

---

## [Editor Report · Decision Letter 1]

12 Mar 2026

Inoculation with indigenous nitrogen-fixers enhances seedling growth and nutrient uptake in a greenhouse bioassay

PONE-D-25-64029R1

Dear Dr. Quoreshi,

We’re pleased to inform you that your manuscript has been judged scientifically suitable for publication and will be formally accepted for publication once it meets all outstanding technical requirements.

Kind regards,

Vishal Tripathi, Ph.D.

Academic Editor

PLOS One
---

## [Editor Report · Acceptance letter]

PONE-D-25-64029R1

PLOS One

Dear Dr. Quoreshi,

I'm pleased to inform you that your manuscript has been deemed suitable for publication in PLOS One. Congratulations! Your manuscript is now being handed over to our production team.

Kind regards,

on behalf of

Dr. Vishal Tripathi

Academic Editor

PLOS One